# Evaluation of the Difference in the Content of Essential and Non-Essential Elements in Wild Boar and Swine Tissues Sampled in the Same Area of Northern Italy

**DOI:** 10.3390/ani14060827

**Published:** 2024-03-07

**Authors:** Susanna Draghi, Michele Spinelli, Carolina Fontanarosa, Giulio Curone, Angela Amoresano, Elisabetta Pignoli, Petra Cagnardi, Daniele Vigo, Francesco Arioli, Stefano Materazzi, Roberta Risoluti, Federica Di Cesare

**Affiliations:** 1Department of Veterinary Medicine and Animal Sciences, University of Milan, Via dell’Università 6, 26900 Lodi, Italy; susanna.draghi@unimi.it (S.D.); daniele.vigo@unimi.it (D.V.); francesco.arioli@unimi.it (F.A.); federica.dicesare@unimi.it (F.D.C.); 2Department of Chemical Sciences, University of Naples Federico II, Via Cintia 21, 80126 Napoli, Italy; michele.spinelli@unina.it (M.S.); carolina.fontanarosa@unina.it (C.F.); angela.amoresano@unina.it (A.A.); epignoli@gmail.com (E.P.); 3I.N.B.B., Istituto Nazionale Biostrutture e Biosistemi, 00136 Roma, Italy; 4Department of Chemistry, Sapienza University of Rome, Piazzale Aldo Moro 5, 00185 Rome, Italy; stefano.materazzi@uniroma1.it (S.M.); roberta.risoluti@uniroma1.it (R.R.)

**Keywords:** eco-toxicology, environmental toxicology, biomonitoring, bioindicators, trace elements

## Abstract

**Simple Summary:**

Trace elements include both essential and non-essential elements, some of which are pollutants, while others occur naturally in the environment. Imbalances among these elements pose health risks. The wild boar serves as a valuable bioindicator, sharing many behavioral traits with the domestic pig, especially with those extensively bred. This study noted minor geographical variations and observed higher levels of specific trace elements in young boars based on age differences. Although sex had no significant impact, distinctions between wild boars and swine highlighted the dietary and environmental influences on element concentrations, underscoring the wild boar’s role as a bioindicator of environmental elements.

**Abstract:**

This study aimed to investigate the exposure of wild boars and swine from semi-extensive farms in the same area to essential and non-essential elements, measuring their concentration in liver and muscle. Furthermore, the study explored the influence of factors such as sex, age, and the sampling location on wild boars. Higher liver element concentrations were observed in both wild boars and swine. Geographical comparisons revealed minor differences. Young wild boars showed significantly higher Cu, Se, Cd, and Cr levels, while older subjects exhibited elevated Mn levels, reflecting age-related element absorption variations. No significant sex-based variations were noted. Comparing wild boars to swine, wild boars had more non-essential elements due to their foraging behavior and a larger home range. Conversely, swine exhibited a greater prevalence of essential elements, potentially resulting from dietary supplementation.

## 1. Introduction

The essential and non-essential elements, also known as trace elements, are classified based on their physiological functions and are regularly found in the environment because of their natural occurrence in the soil or due to anthropogenic activities such as industry, agriculture and transportation [1]. Thus, they can easily enter the food chain, making human, livestock and wild animals daily exposed, and it is widely recognized that an imbalance between them leads to health issues [2].

Elements such as copper (Cu), iron (Fe), manganese (Mn), molybdenum (Mo), cobalt (Co), nickel (Ni), selenium (Se) and zinc (Zn) are considered as essential since they are directly involved in several physiological functions: Zn, Se and Cu are implicated in immune functions, and Se is also important for growth and fertility [3]. Copper is essential for iron metabolism and antioxidant defenses; its excessive intake causes toxicity and its deficiency can impair immune functions and cause anemia [4]. Until a few years ago, Zn and Cu were regularly supplemented in pigs and poultry farming; symptoms caused by excessive Zn supplementation have been reported in swine and include a reduced rate of gain, reduced feed efficiency, depression, and lameness [5]. Moreover, high dietary Zn increases metallothionein synthesis which, at the intestinal level, preferentially binds Cu, sequestering it and leading to Cu deficiency [6].

Molybdenum, considered an essential trace element, protects against Cu and mercury (Hg) poisoning. While low Mo dietary intakes have not induced nutritional deficiency, high Mo intakes have led to secondary Cu deficiency [7].

Among the essential trace elements, Fe is one of the most important; indeed, it serves as a cofactor of a lot of hematoproteins [8]. While Fe deficiency results in anemia and is widely diffused and recognized, Fe overload is infrequent and is mainly caused by genetic pathologies [9].

Cobalt (Co) is essential for rumen microorganisms in cattle, forming vitamin B12. Co deficiency decreases serum vitamin B12, affecting the immune system. Cobalt toxicosis is rare, and is primarily caused by excessive supplementations, though excessive amounts are classified as carcinogenic [10].

Manganese is an important inorganic dietary component involved in the synthesis and activation of many enzymes required for the metabolism of lipids and glucose [11]; in pigs, Mn deficiency causes symptoms related to slow or impaired growth, abnormal metabolism and low fertility [12]. In domestic animals, chronic Mn toxicity causes Fe deficiency due to impaired Fe transportation [13]. Non-essential and potentially toxic elements such as aluminum (Al) and arsenic (As) can interact and influence biochemical pathways. Chromium (Cr), with both biological functions and clinical problems at excessive concentrations, affects nucleic acid structure, immune response, growth, and lipid metabolism. Paradoxically, it appears to promote growth in some ruminants when administered at low concentrations [14].

Non-essential elements like cadmium (Cd) and lead (Pb) have no physiological functions in the body of mammals and even at low concentrations can cause adverse health effects [15,16].

Little is still known about elements such as vanadium (V) and thallium (Tl). Both metals have the ability to generate reactive oxygen species (ROS) and cause clinical syndromes, which include peripheral neuropathy and static and dynamic ataxia [17,18]; conversely, studies have demonstrated that V has anti-diabetic effects and pharmacological activity in the treatment of parasitic diseases, tumors and bacterial and viral infections [19].

Considering the health effects caused by imbalances of essential and non-essential elements, it is crucial to apply methods to identify and quantify their presence in the environment. Studies have shown that the quantification of hazardous substances in the air, water and soil is not sufficient to assess the health risks for humans and animals [20]. The use of living and deceased organisms is currently becoming an established method in biotesting, bioindicating and biomonitoring the environmental presence of organic and inorganic pollutants [21,22,23,24,25]. In particular, due to their foraging behavior, which is strongly bounded with the natural environment, wild animals are exposed to pollutants and could be subjected to deficiencies or excesses of essential elements, meaning that they could represent suitable bioindicators [26].

The wild boar (*Sus scrofa*) is an ubiquitous rooting omnivore present in a variety of ecosystems [27]. Its feeding behavior includes all types of organic matter and, often, inorganic materials such as mud, stones, rubbish and plastics [28]. Research has shown that wild boars are opportunistic feeders; indeed, their diet varies depending on the region they inhabit and the season. For example, a study conducted in Spain found that wild boars consumed a higher proportion of acorns and other fruits during the autumn, while in spring and summer, their diet was dominated by herbaceous plants and roots [28]. Depending on feed availability, they can also adapt their diet to include human-made food sources such as crops and livestock feed [29]. Wild boars are frequently exposed to trace elements in soil due to their opportunistic feeding behavior. These elements accumulate in their tissues, making wild boars valuable bioindicators for assessing local environmental pollution [30]. The domestic swine belongs to the same species as wild boar but is a different subspecies (*Sus scrofa domestica*) [31]. These two subspecies have very similar physiological characteristics and, in the case of swine bred in extensive and semi-extensive farms, similar feeding behavior. The main differences between them are that the swine diet is provided by a controlled feed supplementation, and their home ranges are reduced to fenced areas.

The aim of this study was to investigate, in Northern Italy, the exposure of wild boars and swine bred in semi-extensive farms to essential and non-essential elements by measuring their concentrations in liver and muscle by inductively coupled plasma–mass spectrometry (ICP-MS). In wild boars, the influence of sex, age and the sampling area on their tissue concentrations was investigated. Moreover, comparisons between the essential and non-essential element concentrations in swine and wild boar tissues were made to understand the influences of home ranges and controlled diet.

## 2. Materials and Methods

### 2.1. Sample Collection

For this study, samples were collected from 80 animals (*n* = 40 swine; *n* = 40 wild boar) after obtaining approval from the Ethical Committee (Animal Welfare Organisation of Milan University; authorization number n°26_2022). Firstly, 100 g of liver and 100 g of muscle (Longissimus lumborum et thoracis) were collected from each animal in polyethylene sterile bags, refrigerated at 4 °C, transported to the laboratory, and immediately frozen at −20 °C for further analysis. The liver and muscle samples from wild boar were collected during routine slaughtering procedures at hunting meat processing plants as part of regular wild boar hunting activities from October to December 2021. Morphometric measures were recorded, and the age of the animals was estimated through dental eruption and erosion. Specific criteria for selecting animals for sampling were not applicable, as established killing and monitoring plans for hunting activities provided indications regarding the animals to be killed, their age, and sex. Sampling was conducted in two different provinces of Northern Italy: the Oltrepò Apennines area of Pavia (PV) and the Apennines area of Piacenza (PC) (see Figure 1). The PV area spans 2122 hectares, while the PC area spans 2364 hectares. These two areas are well divided by a natural barrier represented by a portion of the Apennines, with altitudes ranging from 300 to 1300 m above sea level. Both areas are contiguous, non-industrialized, and lack highways, but they differ in agricultural and anthropogenic activity. As depicted in Figure 1, the PV area features larger woods and pastures, with agricultural activity mainly concentrated in the lower regions where wheat, barley, and alfalfa are cultivated. The higher regions are predominantly wooded, consisting mainly of oaks, beeches, and pines. In the PC area, there is more agricultural activity, including the cultivation of wheat, barley, and alfalfa, and several extensive cattle farms are present. Wild boars were grouped according to sex (10 males and 10 females in each area) and age (10 adults, aged >2 years old, and 10 young, aged <2 years old, in each area); the age distribution of the animals is detailed in Appendix A.

In the case of swine, the sampling procedures were conducted at the slaughterhouse under the supervision of the public veterinary service during the same period as that for wild boars. The subjects selected for sampling were chosen randomly from homogeneous groups based on gender and age, and originated from three farms in the PV area. The swine were raised semi-extensively, with the animals allowed to roam freely within fenced areas of approximately two hectares during the day and confined to pigsties at night. The swine were fed a daily ration, as outlined in Appendix A, that was provided every evening. Categorization based on sex and age was not possible for swine, as they were slaughtered for the production of salami, which requires animals of standard age and sex. Categorization based on the sampling area was avoided due to the restricted home ranges of semi-extensive breeding.

### 2.2. Essential and Non-Essential Elements Analysis with ICP-MS

Cobalt (Co), copper (Cu), iron (Fe), manganese (Mn), molybdenum (Mo), nickel (Ni), selenium (Se), zin (Zn), aluminum (Al), antimony (Sb), arsenic (As), barium (Ba), beryllium (Be), cadmium (Cd), chromium (Cr), lithium (Li), lead (Pb), thallium (Tl), vanadium (V) liver and muscle concentrations were quantified using ICP-MS (Agilent 7700, Agilent Technologies, Santa Clara, CA, USA).

#### 2.2.1. Preparation of Standard Solutions and Method Validation

Standard solutions of proper metals were prepared in 3% Nitric Acid at five different concentrations, (0, 1, 10, 50 and 100 μg L^−1^). Concentrated nitric (UpA) and hydrochloric (UpA) acids and certified stock standard solutions were purchased from Romil (ROMIL Ltd., Cambridge, UK).

For each standard, an analysis was performed at different concentrations in a linearity range of 0 to 100 μg/L. The calibration curves for 5 of the 25 metals monitored are shown in the Appendix A: Cr, As, Se, Cd, and Pb (Appendix A).

#### 2.2.2. Sample Treatment

At this stage, 10 g of meat from the wild boar and swine was ground using liquid nitrogen and then 0.25 g was transferred into a 7 mL Pyrex and digested using 4 mL of aqua regia. The reaction was conducted for 16 h at 90 °C. Then, the samples were filtered through 10 µm filter paper, diluted to 25 mL in milli-Q water and transferred into an ICP-MS vial for the analysis. The metal concentrations were measured with three replicates. The measurement was performed with an Agilent 7700 ICP-MS instrument (Agilent Technologies, Santa Clara, CA, USA) equipped with a frequency-matching radio frequency (RF) generator and 3rd generation Octapole Reaction System operating with helium gas in ORF. The following parameters were used: RF power: 1550 W, plasma gas flow: 14 L/min; carrier gas flow: 0.99 L min^−1^; He gas flow: 4.5 mL min^−1^. As an internal standard, 103 Rh was used (final concentration: 50 μg/L).

### 2.3. Statistical Analysis

The statistical analysis was performed with the software Jamovi 2.3.17. For each matrix (liver and muscle), the samples were categorized based on species; the matrices obtained from wild boar were also categorized for sex (male/female), age (young/adult) and area (PV/PC). The preliminary statistical evaluation was performed using the Shapiro–Wilk test, which revealed that the data were not normally distributed. The difference between groups was evaluated by the unpaired Mann–Whitney test. The differences between groups were considered statistically significant when the *p*-value was ≤0.05.

## 3. Results

### 3.1. Detected Concentrations of Essential and Non-Essential Elements in Muscle and Liver of Wild Boar and Swine

Table 1 presents the identified concentrations of essential and non-essential elements in the liver and muscle tissues of wild boar. Similarly, Table 2 outlines the detected concentrations of these elements in the liver and muscle tissues of swine.

### 3.2. Comparison between Concentrations of Essential and Non-Essential Elements in Wild Boar Tissues According to the Two Different Areas

The comparison of the essential and non-essential element concentrations in wild boar tissues from two areas is outlined in Table 3, as well as Appendix A. In terms of liver samples, the median concentrations showed similarity between the two areas for both essential and non-essential elements, with no statistically significant differences observed. In muscle, statistically significant differences were highlighted only for Co, Ni, Se, Zn, with higher concentrations in the PV area.

### 3.3. Comparison between Concentrations of Essential and Non-Essential Elements in Wild Boar Tissues According to Different Age Classes

The comparison of concentrations between age classes in liver and muscle are presented in Table 4, Appendix A. In liver, the concentrations of essential and non-essential elements did not show any significant difference between age classes.

In muscle, Cu, Mn, Se, Cd and Cr showed statistically significant differences. Only Mn was higher in adult wild boars, whereas the other elements were significantly higher in young animals.

### 3.4. Comparison between Mean Concentrations of Essential and Non-Essential Elements in Wild Boar Tissues According to Sex

In Table 5, a comparison between the concentrations of essential and non-essential elements in male and female wild boars is shown. The mean concentration, percentiles, and minimum and maximum levels of essential and non-essential elements in muscle and liver are reported in Appendix A. In liver, only As was significantly higher in male subjects. No significant differences were detected in muscle between males and females.

### 3.5. Comparison between Concentrations of Essential and Non-Essential Elements in Tissues of the Two Species

In Table 6, a comparison between the liver and muscle content of essential and non-essential elements in wild boar and swine is shown. The mean concentrations, percentiles, and minimum and maximum levels are reported in Appendix A. Concerning essential elements, Cu, Fe, Mo, Se, and Zn were higher in the liver of swine.

The non-essential elements As, Cd, Pb and Tl were significantly higher in the liver of wild boar, while V was significantly higher in the liver of swine.

Also, for muscle, statistically significant differences between the two species were reported. Cobalt, Mn, Ni, Al, Ba, Cr, Pb and V were significantly higher in the muscle of wild boar. Copper, Mo, Se, Zn were significantly higher in the muscle of swine.

## 4. Discussion

### 4.1. Tissue Concentration of Essential and Non-Essential Elements in Wild Boar and Swine

Environmental toxicological studies are relevant to controlling the quality of the environment and the presence of pollutants. The examination of tissues and organs from both wild and domesticated animals constitutes a fundamental aspect of these studies. Indeed, any measurable changes in the pollution caused by essential and non-essential elements in the environment are mirrored in animal tissues, as they function as “living sensors” or bioindicators. In contrast to swine, there are a lack of reported baseline levels for essential and non-essential elements in wild boar tissues [32]. Consequently, we refer to EU regulations, which specify safe thresholds for specific non-essential elements in swine tissues for food safety purposes, such as Cd and Pb [33]. These thresholds are derived from the mean content of Cd and Pb quantified in swine meat and offal samples during the national and international monitoring plans, and serve as a point of reference for our assessment of the exposure levels in wild boar. In our research, the Cd concentrations detected in muscle and liver were 0.012 mg⸱kg^−^^1^ and 0.14 mg⸱kg^−^^1^, respectively, and were below the safe limits indicated by EU regulations (0.05 mg⸱kg^−^^1^ ww and 0.5 mg⸱kg^−^^1^ ww, in meat and liver, respectively) (Table 2). The quantified levels of Cd in the muscle and liver of swine (0.02 mg⸱kg^−^^1^ ww and 0.05 mg⸱kg^−^^1^ ww, respectively) were also below these EU limits (Table 2).

In EU regulations, the reported safe levels for Pb are 0.10 mg⸱kg^−^^1^ ww and 0.15 mg⸱kg^−^^1^ ww for meat and offal [33], respectively. Its detected concentrations in the meat and liver of wild boar were 0.049 mg⸱kg^−^^1^ ww and 0.053 mg⸱kg^−^^1^ ww, and 0.041 mg⸱kg^−^^1^ ww in both the meat and liver of swine; they were thus below these European thresholds (Table 1).

The EU regulations report the safe levels of As only for some foodstuffs, such as rice, salt, baby food, etc., but not for meat [33]. Our research revealed that the As levels in wild boar liver and muscle were 0.014 mg⸱kg^−^^1^ ww and 0.008 mg⸱kg^−^^1^ ww, respectively, and in swine both matrices exhibited a concentration of 0.006 mg⸱kg^−^^1^ ww (Table 1 and Table 2). Our findings align closely with the As concentration detected in swine meat in Croatia, which was 0.015 mg⸱kg^−^^1^ ww [34]. In contrast, a study from Hungary [35] reported As concentrations below the limit of quantification (LOQ) of 0.5 mg⸱kg^−^^1^ ww. Furthermore, in Slovakia, Piskorovà et al. [36] found As concentrations of 0.31 mg⸱kg^−^^1^ ww in wild boar muscle. The differences between these studies could be explained not only by differences in the contamination of the regions due to anthropogenic activity, but also by differences in the geological characteristics of the areas. Indeed, As is naturally present in rocks and sediments [37].

In a study conducted by Pilarczyk et al. [38] in Western Ukraine on the contents of essential and non-essential elements in the muscle of wild boar, roe deer and hares, the concentrations of Fe, Cu and Zn were similar to the concentrations detected in the muscle of our wild boar. Generally, in our study, the concentrations of essential elements in samples from swine were higher than those reported for wild animals. When comparing the values we found in swine muscle with those reported in the same species and matrix by Bilandžic’et al. [34], our values were higher, especially for Cu, Fe, Mn, Mo, Se and Zn. It is known that differences in the contents of essential elements in meat are a direct consequence of the animal feed composition, age at slaughtering, breeding techniques and geographical conditions of the breeding area [39].

Compared to the study of Pilarczyk et al. [38], our results for the other elements, such as Al, Ni, Cd and Pb, were lower for both the wild boar and swine tissue concentrations. Our lower concentrations can be attributed to the fact that these elements are primarily derived from anthropogenic activities, and that the Lviv region in Ukraine, studied by Pilarczyk et al., exhibited much more intensive agricultural activity and higher levels of anthropization.

### 4.2. Comparison between Concentrations of Essential and Non-Essential Elements in Tissues of Wild Boar According to the Two Different Areas

Due to anthropological pressure, animals are increasingly being exposed to harmful effects caused by compounds originating from human actions. As public awareness of environmental pollution continues to grow, it becomes crucial to regularly monitor the presence of pollutants in the environment. This monitoring is necessary to ensure that pollutant levels remain within safe and legally recommended thresholds [40]. Some essential and non-essential elements are naturally present in ecosystems, while others are introduced into the environment via animal farming, agricultural activities, urbanization and industrial activity [41]. Given the ubiquitous presence of these elements, it is important to conduct biomonitoring not only in industrialized regions, but also in natural and rural ecosystems located away from emission sources [42,43]. As reported in the results, a panel of elements was quantified in the tissues of wild boar hunted in two adjacent rural areas with different agricultural land uses. Regarding the comparison of the concentrations found in the liver, no statistically significant differences were identified between the groups (Table 3). This lack of significance could be attributed to the similarity in the soil composition, including mineral content, organic matter, clay, silt, and sand distribution, between the two studied areas, as documented in geological maps produced by ISPRA [44]. The texture of the soil and its pH are closely related to its ability to retain substances such as macro- and micro-nutrients. Indeed, soil that has a high percentage of stones made up of large particles has increased nutrient leaching. Conversely, soils with no or a small skeleton will be able to retain more nutrients [45]. Also, the different uses of agricultural land in the two areas do not seem to influence the content of essential and non-essential elements in the tissues of wild boars. In the PC area, alfalfa and cereal crops are cultivated to a greater extent; these crops require phosphoric, potassic and nitrogenous fertilization. This leads to a lack of a substantial increase in the essential and non-essential elements quantified in this study because they are present only as trace elements in the fertilizers used regularly in the area [46,47]. In the comparison of the concentrations of essential elements in muscle between the areas, Co, Ni, Se and Zn were significantly higher in the PV area. Cobalt, an inactive form of vitamin B12, is stored in the muscle [48]; Se has its peak concentration in liver 24 h after exposure; therefore, if the exposure is not recent, it may not be evident in the liver [49]. Zn, after reaching the liver, re-enters the blood circulation to reach the tissues, including the muscle, to become part of metallothionein production at the cellular level [49]. The different tropism of these three elements leads to speculation about different patterns of exposure; their higher presence in muscle might be due to chronic exposure and the storing characteristics of these elements. For example, due to the home ranges of the wild boar, the exposure could have occurred prior to the arrival of the animals in the area. Concerning Ni, its biological role, except for being contained in some enzymes, and its tropism have not been defined yet [50]. This is why the most plausible explanation for its greater presence in the muscle of wild boars from the PV area could be due to a greater presence of this element in the area.

### 4.3. Comparison between Concentrations of Essential and Non-Essential Elements in Tissues of Wild Boar According to the Two Different Age Classes

In general, the accumulation of elements in tissues and organs and liver detoxification activity can vary between young and old animals due to age-related physiological changes and differences in metabolic processes [51]. Despite the normal reduction in liver detoxifying activity associated with aging [52], the comparison between the concentration of elements in the liver of wild boar between age classes did not result in statistically significant differences (Table 4). This could be attributed to the age class division. Indeed, the ages in the group of young animals (<2 y.o.) included wild boar with an age that ranged between 6 months and 2 years old (y.o.); meanwhile, the group of old animals (>2 y.o.) included wild boar with an estimated age that ranged from 2 to 6 y.o., leading to larger standard deviations and a lack of significance (Appendix A). Moreover, most of the adult animals were aged between 2 and 4 y.o., with a mean age of 3.5 y.o., meaning that the plausible detoxifying activity was not yet reduced. Meanwhile, in the muscle of wild boars, statistically significant differences were observed. Copper, Se, Cd and Cr were significantly higher in young animals, whereas Mn was significantly higher in wild boars aged >2 y.o. Concerning Cu, the result can be explained from a physiological point of view, as a difference in terms of absorption between young and adult animals is reported in the literature; hence, the absorption ranges between 5 and 10% of the Cu ingested through the diet in adult animals, whereas young animals are able to absorb from 15 to 30% [48]. These differences are due to several factors, including the following: the higher level of gastric acid secretion in young animals, which increases the Cu absorption [53,54], and the gut microbiota, which may promote the reduction of Cu into a more absorbable form, facilitating its uptake from the intestinal cells [55].

### 4.4. Comparison between Concentrations of Essential and Non-Essential Elements in Tissues of Wild Boar According to Sex

Our findings show a lack of statistically significant differences between males and females in the contents of essential and non-essential elements in the tissues of wild boar; except for As, which was significantly higher in the liver of males (Table 5). The influence of sex on the bioavailability, the transfer and the effects of contaminants and other elements has been evaluated in several studies with different and sometimes conflicting results [56]. Generally, the contents of essential and non-essential elements can be influenced by factors such as the metabolic rate, hormonal pattern and reproductive state [57]. Contrary to our results, a study by Kasprzyk et al. on the content of micro and macro elements in the liver of wild boars reported significant differences between sexes [58]. Also, in different studies on roe deer muscle, liver and hair, differences between the sexes in the contents of essential and non-essential elements were highlighted [37,59]. A possible explanation for the lack of statistically significant sex-related differences in our study is that, unlike the roe deer in other studies, whose females, due to the hunting season, could be in lactation or diapause, the female wild boars used in this study were neither pregnant nor lactating; these are physiological states that are known to greatly mobilize both essential and non-essential elements [60]. Contemporarily, other studies on wild boar liver [61] and red fox liver and kidney [62] were in agreement with our results, showing no sex-related differences in the contents of essential and non-essential elements in tissues. The content of As in the liver of male wild boar was significantly higher than that in females; this finding is in agreement with studies on both animals and humans showning that females present a greater capacity to methylate As and reduce free As and its related toxicity [63,64].

Although animal exposure may be gender-related, it appears that many other factors are capable of influencing the bioaccumulation of non-essential elements and the storage of essential elements.

### 4.5. Comparison between Concentrations of Essential and Non-Essential Elements in Tissues of Wild Boar and Swine

As expected, statistically significant differences between the content of essential and non-essential elements in the tissues of swine and wild boar were detected (Table 6). In general, swine showed significantly higher concentrations of essential elements in both the muscle and liver compared to wild boar, which showed significantly higher concentrations of non-essential elements in their tissues. Even if swine and wild boars are mammals that belong to the same species (*Sus scrofa*), they have different physiological characteristics and feeding behaviors, mainly due to domestication [65]. Domesticated pigs have been bred for specific traits such as rapid growth and high meat production, with the use of selected feedstuff and nutrients to achieve these goals. Swine diets, indeed, provide various supplementations and are specific to the productive and reproductive period, varying, e.g., between weaning, fattening, and gestation. Here, Cu, Fe, Mo, Se, and Zn were significantly higher in the tissues of swine than in those of wild boar. The higher content is probably due to the different diet; indeed, in pig farming, mineral supplements are generally added to the feed [66]. On the contrary, non-essential elements were significantly higher in the tissues of wild boar. This result is probably related to the intrinsic characteristics that wild boar have: larger home ranges, which increase the probability of contact with these elements, and a longer lifespan, which leads to higher bioaccumulation and an omnivorous feeding behavior strongly bound with the natural environment [67].

The differences between the concentrations of trace elements in swine and wild boar tissue may also be related to differences in the gut microbiota. Gut bacterial communities are able to influence the absorption of both essential and non-essential elements and, vice versa, the elements are able to modulate the microbiota [68]. It is widely known for several species, such as rabbits [69], chickens [70], cows [71], that the gut microbiota is largely influenced by the feeding regimen. For example, studies have demonstrated that bacteria such as *Lactobacillus johnsonii* and *Lactobacillus reuteri* have the ability to modulate Fe homeostasis [72]; meanwhile, the effects of Fe supplementation are still little known; in general, Fe is not considered a limiting element in today’s animal production, and some research has found that Fe supplementation increases the presence of beneficial microbiota [73]. Copper and Zn supplementation in pig farming improves growth performance by promoting intestinal health and mucosal integrity and decreasing the relative abundance of potential pathogens such as Enterobacter, Escherichia, and Streptococcus [74,75]. Recent findings have also demonstrated that Zn is able to prevent the formation of biofilms in bacteria such as *A. pleuropneumoniae*, *Salmonella typhymurium* and *Haemophilus parasius.* [76] In chickens infected with *Salmonella typhymurium*, beneficial bacteria such as *Lactobacillaceae* and *Bifidobacteria* colonized and there was a reduction in *Salmonella* and *E. coli* after of the content of Mn in the diet was increased [77]. Moreover, in cases of exposure to non-essential elements, some bacterial genera have the ability to detoxify them; examples are Bacteroides and Alistipes, which carry As resistance genes that are able to methylate this element [78]. Conversely, during exposure to Pb, Bacteroidetes and Proteobacteria are enriched [79], and exposure to a mixture of non-essential elements such as Cd, Cr, and Pb causes alterations in bacterial functions that can lead to changes in their metabolites and consequently to disease conditions [80]. Diet is not the unique factor implicated in the modulation of microbiota. Indeed, gut microbiota diversity is also a result of co-evolution between the host and the microorganisms [81]; a study conducted by Wei et al. [82] showed that domestic pigs have a significantly higher abundance of some species of Lactobacillus, and reduced relative abundances of Bifidobacterium and Methanococcaceae. Moreover, in wild boar, some metabolic functions of the gut microbiota are increased, such as “environmental adaptation”, “disease resistance” and “immune function” [66].

## 5. Conclusions

In conclusion, this study illustrates the relationships between animal physiology, environmental factors, and element accumulation. Moreover, the differences observed between swine and wild boar highlight the importance of considering species-specific behaviors and dietary habits when assessing the impacts of pollutants on wildlife and livestock. In a one health perspective approach, considering that the consumption of wild boar meat is negligible, the wild boar assumes the role of a bio-indicator, providing indications of the elements present in the area. Additional research, quantifying the concentrations of elements in soil, water, and feed, may be necessary to establish correlations with tissue concentrations and gain a deeper insight into the factors influencing exposure.

## Figures and Tables

**Figure 1 animals-14-00827-f001:**
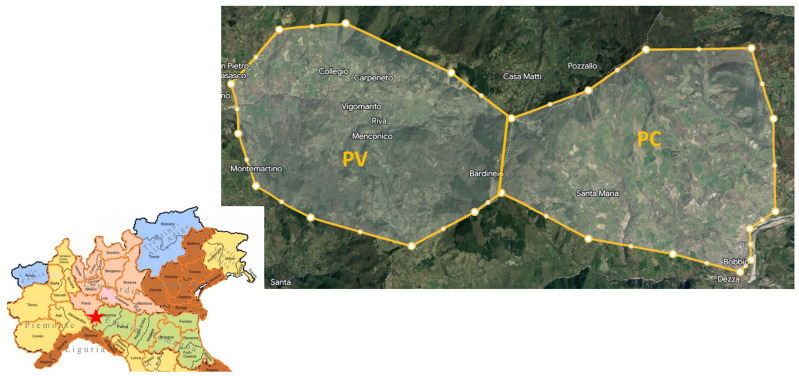
Map of the sampling area. PV: Oltrepo’ Appenines of Pavia; PC: Val Trebbia Appenines of Piacenza; The red star represents the location of the sampling area on the map of northern Italy.

**Table 1 animals-14-00827-t001:** Mean ± SD, median with percentile and ranges of detected concentrations of essential and non-essential elements in liver and muscle of wild boar. Values are reported in µg⸱kg^−1^ ww.

Element	Wild Boar (*n* = 40)					
	Liver (*n* = 40)			Muscle (*n* = 40)		
	Mean ± SD	Median (P25; P75)	Min–Max	Mean ± SD	Median (P25; P75)	Min–Max
*Co*	27.05 ± 4.42	25.33 (23.67; 30.15)	21.43–36.63	7.37 ± 5.53	5.19 (3.91; 5.86)	3.07–20.54
*Cu*	3424.16 ± 569.92	3495.93 (2906.65; 3797.72)	2393.55–4606.05	1740.77 ± 497.28	1572.22 (1429.93; 1786.96)	1155.87–2909.51
*Fe*	218,701.16 ± 78,400.76	206,287.79 (160,623.34; 299,503.53)	109,920.34–356,189.4	30,066.85 ± 8479.75	28,726.03 (22,563.59; 34,355.85)	20,134.93–47,678.89
*Mn*	2374.59 ± 835.9	2301.61 (1582.97; 3088.43)	1396.41–3917.74	607.92 ± 332.26	562.58 (305.99; 931.27)	157.88–1126.62
*Mo*	724.90 ± 192.66	626.64 (559.68; 909.07)	481.11–1039.25	9.95 ± 3.76	11.04 (8.39; 12.41)	0–14.18
*Ni*	27.61 ± 8.16	26.25 (21.35; 35.23)	15.59–41.06	70.89 ± 35.27	71.78 (57.05; 94.83)	17.98–140.74
*Se*	239.06 ± 47.47	245.12 (208.26; 276.96)	148.57–309.41	101.73 ± 62.24	84.80 (66.12; 99.84)	34–235.83
*Zn*	31,236.67 ± 3644.436	30,197.55 (28,716.08; 32,883.00)	25,470.24–40,956.24	30,725.82 ± 16,568.12	21,048.99 (18,542.67; 47,944.67)	15,558.35–65,909.36
*Al*	4055.12 ± 585.28	3997.19 (3650.7; 4275.25)	3155.220–5814.65	5343.46 ± 2089.05	4160.85 (3411.58; 7398.73)	2943.25–8990.14
*As*	14.49 ± 9.250	10.71 (7.81; 14.87)	7.06–34.65	8.76 ± 1.48	8.44 (7.56; 9.74)	6.75–11.84
*Ba*	73.00 ± 20.69	68.69 (53.4; 94.38)	49.83–105.29	136.41 ± 92.82	108.53 (63.73; 148.91)	53.88–358.98
*Cd*	138.67 ± 109.89	53.87 (48.35; 243.55)	38.52–313.41	12.6 ± 4.57	10.43 (8.87; 15.80)	7.16–21.59
*Cr*	53.39 ± 24.19	44.23 (39.83; 48.61)	35.91–122.78	137.29 ± 96.16	99.02 (64.94; 138)	45.25–344.67
*Pb*	53.75 ± 11.06	54.09 (42.79; 61.06)	38.94–71.96	49.56 ± 8.09	49.60 (43.13; 55.72)	35.76–65.91
*Tl*	5.46 ± 3.99	5.33 (1.94; 9.80)	0.690–11.55	0.06 ± 0.17	0 (0; 0)	0–0.61
*V*	46.14 ± 9.06	44.45 (38.96; 51.55)	33.56–68.9	72.7 ± 9.15	72.18 (65.58; 79.67)	54.96–97.24

**Table 2 animals-14-00827-t002:** Mean ± SD, median with percentile and ranges of detected concentrations of essential and non-essential elements in liver and muscle of swine. Values are reported in µg⸱kg^−1^ ww. If the element was not found, the column will show “N.D.” for not detected.

Element	Swine (*n* = 40)					
	Liver (*n* = 40)			Muscle (*n* = 40)		
	Mean ± SD	Median (P25; P75)	Min–Max	Mean ± SD	Median (P25; P75)	Min–Max
*Co*	28.25 ± 8.16	27.91 (23.882; 34.16)	13.79–44.65	3.3 ± 0.99	3.26 (2.402; 3.98)	2.07–5.82
*Cu*	16,700.72 ± 8969.23	16820 (9115.433; 18,882.5)	7279.59–34,576.3	2192.88 ± 267.98	2234.89 (1958.62; 2421.32)	1696.51–2661.15
*Fe*	336,705.3 ± 95,226.41	318,963 (282,535.73; 390318)	186,823.61–572,838.78	30,023.03 ± 5011.51	29,857.6 (26,520.222; 33,008.9)	20,760.52–42414
*Mn*	2634.62 ± 977.11	2657.7 (1656.914; 3181.25)	1315.67–4737.56	344.42 ± 72.71	353.97 (283.923; 402.32)	188.29–445.62
*Mo*	2054.06 ± 307.38	2076.3 (1927.266; 2198.3)	1421.12–2718.25	32.14 ± 5.93	32.7 (29.2; 35.49)	18.64–42.71
*Ni*	58.99 ± 57.33	35.05 (13.145; 132.07)	10.55–153.44	22.19 ± 23.45	10.5 (9.352; 12.75)	8.17–71.17
*Se*	320.24 ± 25.86	318.9 (297.485; 341.33)	273.18–371.9	103.71 ± 13.69	102.44 (92.343; 112.35)	84.83–130.56
*Zn*	67,454.11 ± 18,018.53	67,345.9 (51,419.637; 80,832.1)	36,310.74–100,655.95	46,536.27 ± 5740.44	47,079.3 (42,291.388; 50,876.6)	36,327.3–59,087.38
*Al*	12,029.17 ± 9551.85	13,290.7 (2683.48; 18,216.7)	2176.35–28,549.38	3191.31 ± 839.44	2842.53 (2602.878; 3995.05)	2126.67–4965.56
*As*	6.28 ± 1.73	6.31 (5.083; 7.13)	3.30–10.24	6.41 ± 1.47	5.77 (5.13; 7.82)	4.4–8.85
*Ba*	73.94 ± 42.03	75.77 (38.763; 84.54)	32.65–162.66	49.47 ± 17.19	44.83 (35.442; 65.02)	19.92–80.39
*Cd*	48.34 ± 11.25	45.45 (42.146; 51.07)	33.074–75.59	2.49 ± 1.79	1.98 (0.682; 3.24)	0.56–6.1
*Cr*	116.37 ± 127.26	47.8 (6.71; 281.41)	5.52–329.07	25.02 ± 15.67	19.9 (14.027; 39.67)	8.73–54.67
*Pb*	41.17 ± 12.60	44.03 (35.387; 50.52)	17.18–63.36	41.06 ± 39.81	10.59 (7.09; 38.22)	4.97–115.45
*Tl*	N.D.	N.D.	N.D.	N.D.	N.D.	N.D.
*V*	68.4 ± 21.47	66.94 (51.26; 72.46)	35.93–113.46	59.44 ± 7.17	60.58 (53.547; 64.4)	46.39–74.19

**Table 3 animals-14-00827-t003:** Comparison of concentrations in the liver and muscle of wild boars in the two study areas (PV and PC). The data are reported in µg⸱kg^−1^ww. In the “*p*-value” column, only statistically significant *p*-values (p≤0.05) are indicated.

Element	Liver (*n* = 40)			Muscle (*n* = 40)		
	PV (*n* = 20)	PC (*n* = 20)	*p*-Value	PV (*n* = 20)	PC (*n* = 20)	*p*-Value
	Median	Median		Median	Median	
*Co*	26.58	24.25		5.53	4.09	*p* = 0.02
*Cu*	3516.74	3435.96		1574.78	1535.15	
*Fe*	207,533.6	189,854.52		29,112.98	25,517.2	
*Mn*	2441.56	2232.57		624.47	340.34	
*Mo*	769.26	575.11		10.59	11.85	
*Ni*	26.15	26.99		78.82	59.82	*p* = 0.03
*Se*	249.75	240.49		89.57	73.7	*p* = 0.05
*Zn*	29,814.06	31,057.21		23,017.68	18,878.96	*p* = 0.004
*Al*	4054.09	3996.69		4359.30	3961.37	
*As*	13.76	8.77		8.41	8.47	
*Ba*	78.56	55.65		117.89	68.64	
*Cd*	206.31	52		10.04	13.26	
*Cr*	43.42	43.84		99.35	91.1	
*Pb*	48.77	56.75		49.21	50.42	
*Tl*	2.29	5.63		0.00	0	
*V*	47.04	41.56		72.31	71.82	

**Table 4 animals-14-00827-t004:** Comparison of concentrations in the liver and muscle of wild boars in the two age classes (adult >2 years old, and young animals <2 years old). The data are reported in µg⸱kg^−1^ ww. In the “*p*-value” column, only statistically significant *p*-values (p≤0.05) are indicated.

Element	Liver (*n* = 40)			Muscle (*n* = 40)		
	<2 y.o. (*n* = 20)	>2 y.o. (*n* = 20)	*p*-Value	<2 y.o. (*n* = 20)	>2 y.o. (*n* = 20)	*p*-Value
	Median	Median		Median	Median	
*Co*	25.34	25.32		4.24	5.21	
*Cu*	3511.46	3485.72		1599.14	1470.63	*p* = 0.03
*Fe*	189,923.4	211,863.3		23,477.91	30,163.97	
*Mn*	2433.28	1853.24		340.34	725.86	*p* = 0.02
*Mo*	607.01	659.91		11.73	10.08	
*Ni*	25.01	30.2		75.9	68.48	
*Se*	251.88	230.21		89.57	71.06	*p* = 0.02
*Zn*	29,585.43	31,264.43		20,037.37	21,271.08	
*Al*	3976.28	4094.93		3780.52	6379.95	
*As*	8.64	12.9		8.35	8.59	
*Ba*	59.59	77.79		70.07	124.32	
*Cd*	51.76	55		13.52	9.29	*p* = 0.01
*Cr*	42.31	44.71		99.38	65.09	*p* = 0.03
*Pb*	54.93	47.32		48.39	53.17	
*Tl*	5.49	4.96		0	0	
*V*	41.34	47.43		72.88	71.71	

**Table 5 animals-14-00827-t005:** Comparison of concentrations in the liver and muscle of wild boars in the two age classes (*male* and *female*). The data are reported in µg⸱kg^−1^ ww. In the “*p*-value” column, only statistically significant *p*-values (*p* ≤ 0.05) are indicated.

Element	Liver (*n* = 40)			Muscle (*n* = 40)		
	Female (*n* = 20)	Male (*n* = 20)	*p*-Value	Female (*n* = 20)	Male (*n* = 20)	*p*-Value
	Median	Median		Median	Median	
*Co*	24.95	27.14		4.29	5.38	
*Cu*	3496.14	3493.19		1545.97	1599.36	
*Fe*	194,796.2	208,452.6		26,209.71	29,981.99	
*Mn*	2368.74	2076.13		356.83	651.45	
*Mo*	577.57	714.59		11.62	10.51	
*Ni*	26.25	27.81		61.72	78.93	
*Se*	245.12	245.37		84.8	85.77	
*Zn*	29,885.11	31,688.08		19,800.6	22,250.11	
*Al*	3962.66	4057.99		3686.66	5362.61	
*As*	8.43	13.56		8.47	8.44	*p* = 0.015
*Ba*	58.34	81.69		66.45	122.19	
*Cd*	51.88	130.16		13.39	9.9	
*Cr*	44.23	42.79		99.36	96.38	
*Pb*	56.12	46.59		47.75	50.2	
*Tl*	5.43	3.9		0	0	
*V*	41.8	47.89		74.19	71.37	

**Table 6 animals-14-00827-t006:** Comparison of concentrations in the liver and muscle of wild boars and swine. The data are reported in µg⸱kg^−1^ ww. In the “*p*-value” column, only statistically significant *p*-values are indicated.

Element	Liver (*n* = 80)			Muscle (*n* = 80)		
	Swine (*n* = 40)	Wild Boar (*n* = 40)	*p*-Value	Swine (*n* = 40)	Wild Boar (*n* = 40)	*p*-Value
	Median	Median		Median	Median	
Co	27.91	25.33	0.287	3.26	5.19	<0.001
Cu	16,819.97	3495.93	<0.001	2234.89	1572.22	<0.001
Fe	318,962.8	206,287.8	<0.001	29,857.6	28,726.03	0.423
Mn	2657.7	2301.61	0.284	353.97	562.58	0.009
Mo	2076.3	626.64	<0.001	32.7	11.04	<0.001
Ni	35.05	26.25	0.854	10.5	71.78	<0.001
Se	318.91	245.12	<0.001	102.44	84.8	0.002
Zn	66,518.42	30,197.55	<0.001	47,079.32	21,048.99	<0.001
Al	13,290.72	3997.19	0.537	2842.53	4160.85	<0.001
As	6.31	10.71	<0.001	5.77	8.44	<0.001
Ba	75.77	68.69	0.124	44.83	108.53	<0.001
Cd	45.45	53.87	<0.001	1.98	10.43	<0.001
Cr	47.8	44.23	0.27	19.9	99.02	<0.001
Pb	44.03	54.09	<0.001	10.59	49.6	0.079
Tl	N.D.	5.33		N.D.	0	
V	66.94	44.45	<0.001	60.58	72.18	<0.001

## Data Availability

The data presented in this study are available in the article and Appendix A. Further information is available upon request from the corresponding author.

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
