# Peer review of "Evaluation of the Difference in the Content of Essential and Non-Essential Elements in Wild Boar and Swine Tissues Sampled in the Same Area of Northern Italy"

_animals, 2024, doi:10.3390/ani14060827_

Round 1
Reviewer 1 Report
Comments and Suggestions for Authors
Although this study presents relevant results that should be shared with the scientific community, I believe this manuscript needs considerable changes not only in its writing & presentation but also in the perspective and conclusions that can be taken from this work. Moreover, though I am not an English native speaker, I believe the manuscript needs to be revised for English. Some sentences are imprecise or using expressions that are very unformal or poorly structured.
Here I present some of the aspects that really need to be corrected before considering publication.
1) I disagree with the designation of PTE (potentially toxic elements) because I believe this is not scientifically correct. By definition, every element and substance on Earth is potentially toxic depending on the dose, frequency or route of exposure. "Metals and metalloids" (or "metal(loid)s") should be used or, alternatively, "trace elements".
2) L42 Wild boars are swine as well. Pigs are domestic swine, while wold boars are wild swuine, therefore I do not agree with the designation "swine" exclusevely of domestic pigs.
3) L51-53 - The first sentence of the manuscript is very imprecise and, as the authors say in the rest of the paragraph, it does not make any sense for the pollutants that these authors work with. Metal(loid)s normally appear and increase in the Earth crust due to natural events as vulcaninc eruption. However, anthropogenic activities as agricultural, industrial or mining activity also contributes to increase the concentrations of those. This should be rephrased.
4) L128-129 "These two species owned the same physiological characteristics" - this is very imprecise, I understand what the authors mean but it is referred to what? WHy is it improtant for a toxicological assessment?
5) Authors do not refer if their concentrations are expressed in dry weight or wet weight and this is very important, also to compare with other similar studies. Considering the methodology I am assuming wet weight but this should be expressed "μg⸱kg-1 ww"
6) Authors should indicate the obtained value of p when presenting a a statistical result, and not simply say p<0.05.
7) All the text in the results section is repeting the figures. Authors should highlight the main findings but not repeat the whole information.
8) Authors should also analyse and discuss these results under a One Health perspective (human, animal and environmental health). The conclusion regarding the differences between domestic pigs and wild boars is valid but what can be really revealed from your study? What is the risk for human consumers of this meat or for other animals that share the ecosystem with them? Recent wild boar studies in Europe (namely in Spain) have analysed this aspect in deep.
Author Response
Reviewer 1
Although this study presents relevant results that should be shared with the scientific community, I believe this manuscript needs considerable changes not only in its writing & presentation but also in the perspective and conclusions that can be taken from this work. Moreover, though I am not an English native speaker, I believe the manuscript needs to be revised for English. Some sentences are imprecise or using expressions that are very unformal or poorly structured.
Here I present some of the aspects that really need to be corrected before considering publication.
1) I disagree with the designation of PTE (potentially toxic elements) because I believe this is not scientifically correct. By definition, every element and substance on Earth is potentially toxic depending on the dose, frequency or route of exposure. "Metals and metalloids" (or "metal(loid)s") should be used or, alternatively, "trace elements".
We thank the reviewer for the request for clarification. The locution was used on purpose, with the belief that potentially toxic elements is the most scientifically correct way of defining elements that are found in both metallic and ionized states. moreover, IUPAC in one of its reports (Pure Appl. Chem, Vol. 74, No. 5, pp. 793-807, 2002) in which strongly advises against using the term heavy in front of metals, also reminds us not to use even the single word toxic,
“Toxic metal. An imprecise term. The fundamental rule of toxicology (Paracelsus, 1493–1541) is that all substances, including carbon and all other elements and their derivatives, are toxic given a high enough dose. The degree of toxicity of metals varies greatly from metal to metal and from organism to organism. Pure metals are rarely, if ever, very toxic (except as very fine powders, which may be harmful to the lungs from whatever substance they may originate). Toxicity, like essentiality, should be defined by reference to a dose–response curve for the species under consideration. This is another term that has been used loosely to refer to both the element and its compounds”
Certainly, expressing the potential toxicity is indeed accurate. Regarding the term "metalloid," IUPAC's precision is not as explicit, and it appears to treat metalloid and semimetal interchangeably.
Moreover, the use of the term "element" instead of categorizing them as metals stems from the observation that these substances are frequently found in an ionized state. Consequently, many chemists and several articles prefer referring to them as PTEs. We adhere to this definition, as exemplified in some publications (see, for instance, https://doi.org/10.3390/agriculture13091684; https://doi.org/10.1007/s11356-021-14879-2; https://doi.org/10.1016/j.chemosphere.2022.133649; ).
2) L42 Wild boars are swine as well. Pigs are domestic swine, while wold boars are wild swuine, therefore I do not agree with the designation "swine" exclusevely of domestic pigs.
We thank the reviewer for the request for clarification. Wild boars (Sus scrofa), Domestic swine (Sus scrofa domesticus), and Feral pigs (Sus scrofa scrofa) should not be considered synonymous. Wild boars are the progenitors, domestic pigs are the result of domestication of wild boars, and feral pigs are unowned and freely roaming. When making reference to wild pigs, it is important to note that this term denotes feral pigs specifically (source: https://doi.org/10.1016/B978-0-323-85125-1.00111-3).
3) L51-53 - The first sentence of the manuscript is very imprecise and, as the authors say in the rest of the paragraph, it does not make any sense for the pollutants that these authors work with. Metal(loid)s normally appear and increase in the Earth crust due to natural events as vulcaninc eruption. However, anthropogenic activities as agricultural, industrial or mining activity also contributes to increase the concentrations of those. This should be rephrased.
We thank the reviewer and agree with him for this reason the sentence has been removed (Line 51-55) .
4) L128-129 "These two species owned the same physiological characteristics" - this is very imprecise, I understand what the authors mean but it is referred to what? WHy is it improtant for a toxicological assessment?
We thank the reviewer for the request for clarification. The sentence has been revised to enhance clarity. When discussing wild boar (Sus scrofa), domestic pigs (Sus scrofa domesticus), or feral pigs (Sus scrofa scrofa), we are referring to distinct subspecies within the same species. Evolutionarily, these subspecies share strikingly similar physiological characteristics but have adapted to diverse environments through subtle modifications in behavior, and possibly (a speculative assertion) some aspects of their digestive and reproductive systems (Lines 137-139).
5) Authors do not refer if their concentrations are expressed in dry weight or wet weight and this is very important, also to compare with other similar studies. Considering the methodology, I am assuming wet weight but this should be expressed "μg⸱kg-1 ww"
We thank the reviewer for the request for clarification. The concentrations are expressed on wet weight. The wording ww have been added.
6) Authors should indicate the obtained value of p when presenting a a statistical result, and not simply say p<0.05.
The obtained p value has been added for each element, not only in the text but also in graphs.
7) All the text in the results section is repeting the figures. Authors should highlight the main findings but not repeat the whole information.
To enhance readability, we have chosen to include in the text only the concentrations of elements that exhibited statistically significant differences. The comprehensive data can be found solely in the supplementary materials tables.
8) Authors should also analyse and discuss these results under a One Health perspective (human, animal and environmental health). The conclusion regarding the differences between domestic pigs and wild boars is valid but what can be really revealed from your study? What is the risk for human consumers of this meat or for other animals that share the ecosystem with them? Recent wild boar studies in Europe (namely in Spain) have analysed this aspect in deep.
We thank the reviewer for the suggestion. While consumer health is a prominent concern, it's important to note that this study did not involve the calculation of consumer risk. The primary focus of this research was on utilizing wild animals as bioindicators to assess the presence and levels of potentially toxic elements, particularly in areas where semi-extensively reared domestic pigs share pastures with wildlife. This approach allows for the consideration of a reduced risk to consumers by minimizing the use of pastureland by farms. Additionally, the consideration was influenced by the fact that the consumption of wild boar meat is very limited and negligible in the study area. A sentence emphasizing the one health approach was incorporated into the conclusions to better contextualize the objectives of the study.
Reviewer 2 Report
Comments and Suggestions for Authors
This study investigates the concentrations of essential and non-essential elements in the liver and muscle tissues of wild boars and domestic swine. The research focuses on understanding the impact of environmental factors and dietary habits on element accumulation in these animals. The paper's strengths include its comprehensive analysis of a wide range of elements, the comparison between wild and domestic species.
Comments:
1. The Introduction provides details of the function of essential and non-essential elements for the animals. However, these details might be more appropriate in the discussion part. The paper would benefit from a summary of how these findings relate to animal health and potential risks to human consumers, or even add some experimental data to illustrate the animal health of different groups.
2. The methodological approach is thorough, but there is a need for a more precise explanation of animal husbandry, sample collection procedures, animal selection criteria, and statistical analysis methods to enhance reproducibility and clarity. In addition, there is no table showing the formulation of the concentration that used for the swine, and also there is no data to support the different contents of essential and non-essential elements in the soil which should be the major contribution to the differences.
3. The first part of Result (3.1) is about comparing the contents of elements in the Liver and Muscle of each animal. It is obvious that different organs should have different deposition of essential and non-essential elements. This might not be an appropriate comparison, and no proper control group exists. I would suggest including the growth performance of each group in the result.
4. (line 589) The discussion on the role of gut microbiota in element absorption is intriguing but somewhat speculative. More direct evidence or additional research in this area could strengthen these claims.
5. Inclusion of recent studies on similar topics and a more comprehensive review of the literature would provide a better contextual background for the study.
6. The * in each figure is not in the appropriate position. Professional figure software is recommended to re-draw each figure.
Author Response
Reviewer 2
his study investigates the concentrations of essential and non-essential elements in the liver and muscle tissues of wild boars and domestic swine. The research focuses on understanding the impact of environmental factors and dietary habits on element accumulation in these animals. The paper's strengths include its comprehensive analysis of a wide range of elements, the comparison between wild and domestic species.
Comments:
- The Introduction provides details of the function of essential and non-essential elements for the animals. However, these details might be more appropriate in the discussion part. The paper would benefit from a summary of how these findings relate to animal health and potential risks to human consumers, or even add some experimental data to illustrate the animal health of different groups.
We thank the reviewer for the valuable suggestion, regarding the composition of the introductory paragraph, the authors aimed to initially center the introduction around broader aspects of potentially toxic elements (PTEs). Following this initial discussion, the intention was to transition the emphasis towards exploring the utilization of animals as bioindicators for assessing the presence and concentrations of PTEs within the environment. Following the reviewer's advice, we have endeavoured to condense and streamline the introductory section for improved fluency.
- The methodological approach is thorough, but there is a need for a more precise explanation of animal husbandry, sample collection procedures, animal selection criteria, and statistical analysis methods to enhance reproducibility and clarity. In addition, there is no table showing the formulation of the concentration that used for the swine, and also there is no data to support the different contents of essential and non-essential elements in the soil which should be the major contribution to the differences.
We thank the reviewer for the request for clarification. Some details concerning swine husbandry, sample collection procedures, animal selection criteria and statistical analysis have been added in sections 2.1. and 2.3.
The table with the feed formulation was added to the supplementary materials (Table S1).
Due to the extensive size of the two areas from which the animals originate, we have decided not to conduct soil investigations, a decision also made in several articles that use animals as environmental bioindicators for large areas (some examples: doi.org/10.3389/fenvs.2021.794487; doi.org/10.1016/j.chemosphere.2019.125506). This decision is based on the fact that the terrains in both territories exhibit significant variability in terms of composition and stratigraphy. Additionally, wild boars, being a species capable of covering long distances throughout the day and, therefore, able to forage in different zones inside of the two hunting areas, represent, from our perspective, ideal animals for comparing the extensive content of potentially toxic elements in large-scale areas
- The fithispart of Result (3.1) is about comparing the contents of elements in the Liver and Muscle of each animal. It is obvious that different organs should have different deposition of essential and non-essential elements. This might not be an appropriate comparison, and no proper control group exists. I would suggest including the growth performance of each group in the result.
We thank the reviewer for the suggestion. We are aware that the comparison between the two matrices produces an evident outcome due to the differing toxicokinetics of the organs. After a comprehensive literature review, we observed that many authors commonly address this comparison (some examples doi: 10.3390/foods11162530; doi: 10.1080/02652030050034028; https://doi.org/10.1016/j.chemosphere.2019.125506). Furthermore, unlike other studies indicating consistently higher element content in the liver, our examination of wild boar species found elevated levels in the muscle. The resulting discrepancies are thoroughly discussed in the respective paragraph 4.1.
- (line 589) The discussion on the role of gut microbiota in element absorption is intriguing but somewhat speculative. More direct evidence or additional research in this area could strengthen these claims.
We thank the reviewer for the suggestion. In paragraph 4.5, starting from line 601, we have incorporated several instances illustrating the modulation of gut microbiota through the utilization of essential elements.
- Inclusion of recent studies on similar topics and a more comprehensive review of the literature would provide a better contextual background for the study.
We thank the reviewer for the suggestion. Several additional recent studies have been included.
- The * in each figure is not in the appropriate position. Professional figure software is recommended to re-draw each figure.
We thank the reviewer for the suggestion. The software graphpad prism 8 has been used to draw each figure, the significance has now been expressed with both p value and *: p < 0.05; **: p < 0.01; ***: p < 0.001.
Round 2
Reviewer 1 Report
Comments and Suggestions for Authors
From my perspective, although authors have changed their manuscript in very few of the aspects I mentioned in my previous report, a significant number of the authors' answers do not have a correction and there are still issues to solve.
Author Response
We extend our gratitude to Reviewer One for the invaluable suggestions provided throughout the review process, to which we had already responded in previous rounds of revision. The final steps have been overseen by the academic editors. We thank the academic editors for their assistance and guidance throughout the review process. Their expertise and thoughtful suggestions have significantly enhanced the quality and clarity of our manuscript. In response to their feedback, we have diligently implemented all recommended changes, including the removal of Table 1 and the renumbering of subsequent tables for coherence. Furthermore, we have taken the opportunity to refine and restructure certain sections of the materials and methods to ensure greater comprehensibility and accuracy in English.We sincerely appreciate the academic editors for believing in and supporting the concept behind this scientific paper.
Reviewer 2 Report
Comments and Suggestions for Authors
The authors have answered all my questions.
Author Response
We extend our gratitude to Reviewer Two for the invaluable suggestions provided throughout the review process, to which we had already responded in previous rounds of revision. The final steps have been overseen by the academic editors. We thank the academic editors for their assistance and guidance throughout the review process. Their expertise and thoughtful suggestions have significantly enhanced the quality and clarity of our manuscript. In response to their feedback, we have diligently implemented all recommended changes, including the removal of Table 1 and the renumbering of subsequent tables for coherence. Furthermore, we have taken the opportunity to refine and restructure certain sections of the materials and methods to ensure greater comprehensibility and accuracy in English.We sincerely appreciate the academic editors for believing in and supporting the concept behind this scientific paper.